# Basigin drives intracellular accumulation of L-lactate by harvesting protons and substrate anions

**Anna-Lena Köpnick, Annika Jansen, Katharina Geistlinger, Nathan Hugo Epalle, Eric Beitz** ☉ *

Department of Pharmaceutical and Medicinal Chemistry, Christian-Albrechts-University of Kiel, Kiel, Germany

* ebeitz@pharmazie.uni-kiel.de

**Data Availability Statement:** All relevant data are within the manuscript and its Supporting information files.

## Abstract

Transmembrane transport of L-lactate by members of the monocarboxylate transporter family, MCT, is vital in human physiology and a malignancy factor in cancer. Interaction with an accessory protein, typically basigin, is required to deliver the MCT to the plasma membrane. It is unknown whether basigin additionally exerts direct effects on the transmembrane L-lactate transport of MCT1. Here, we show that the presence of basigin leads to an intracellular accumulation of L-lactate 4.5-fold above the substrate/proton concentrations provided by the external buffer. Using basigin truncations we localized the effect to arise from the extracellular Ig-I domain. Identification of surface patches of condensed opposite electrostatic potential, and experimental analysis of charge-affecting Ig-I mutants indicated a bivalent harvesting antenna functionality for both, protons and substrate anions. From these data, and determinations of the cytosolic pH with a fluorescent probe, we conclude that the basigin Ig-I domain drives lactate uptake by locally increasing the proton and substrate concentration at the extracellular MCT entry site. The biophysical properties are physiologically relevant as cell growth on lactate media was strongly promoted in the presence of the Ig-I domain. Lack of the domain due to shedding, or misfolding due to breakage of a stabilizing disulfide bridge reversed the effect. Tumor progression according to classical or reverse Warburg effects depends on the transmembrane L-lactate distribution, and this study shows that the basigin Ig-I domain is a pivotal determinant.

## Introduction

L-Lactate is a key intermediate in the energy metabolism [1]. It is generated together with a proton as the main product of anaerobic glycolysis demanding swift release from the cell to avoid acidification. To other cells, circulating L-lactate serves as a substrate for gluconeogenesis and lipogenesis, or as an energy source fueling the tricarboxylic acid cycle [1]. A century ago, Warburg described a property of various tumors that undergo anaerobic glycolysis and release of L-lactate despite an ample availability of oxygen, termed the Warburg effect [2]. Later, other

**Funding:** This project has received funding from the European Union's Horizon 2020 research and innovation programme under the Marie Skłodowska-Curie grant agreement No. 860592.

**Competing interests:** The authors have declared that no competing interests exist.

**Abbreviations:** BSG, basigin; GFP, green fluorescent protein; Ig-C2, immunoglobulin-like domain C2; Ig-I, immunoglobulin-like domain l; MCT, monocarboxylate transporter; SD medium, synthetic dropout medium.

tumors were found to effectively feed on circulating L-lactate by a reverse Warburg effect [1,3]. Inhibition of transmembrane L-lactate transport is, thus, viewed to bear potential as a novel approach against cancer [4–7].

At least four secondary active monocarboxylate transporters (MCT1-4) of the 14-membered solute carrier family SLC16A mediate bidirectional proton-coupled transport of L-lactate across the plasma membrane [8]. The prototypical, intermediate-affinity MCT1 ($K_m \approx 3.5$ mM) is ubiquitously expressed and mediates both, import and export of L-lactate in a physiological setting, whereas the high-affinity MCT2 ($K_m \approx 0.7$ mM) and the low-affinity MCT4 ($K_m \approx 28$ mM) appear more geared to facilitate L-lactate uptake or release, respectively [9,10]. The recently solved structures of a bacterial MCT homolog, and human MCT1 and MCT2 widely confirmed earlier models on the principal 12 transmembrane spanning topology and alternating access transport mechanism [11–13].

MCTs form associations with one of two proteins of the immunoglobulin superfamily, namely embigin and basigin, that are crucial for intracellular trafficking, with human MCT1 preferring basigin [14]. It is unknown if the basigin association additionally affects L-lactate transport directly. Such modulations are potentially connected to the malignancy of certain types of cancer in which expression of specific basigin isoforms is upregulated [15–17]. The four basigin isoforms consist of a single transmembrane helix and carry one to three immunoglobulin-(Ig)-like domains towards the extracellular N-terminus [15,18–20]. Basigin variant 2 (BSG var2, also termed BSG, CD147 or EMMPRIN; Fig 1A) is the most prevalent isoform and serves as a receptor for integrins, cyclophilins, and for surface proteins of infective agents, such as the malaria parasite *Plasmodium falciparum* [21] and common viruses including the immunodeficiency virus (HIV) [22] and the severe acute respiratory syndrome (SARS) coronavirus (CoV) [23]. Basigin further activates extracellular matrix metalloproteinases [24], and recruits carbonic anhydrase IV to MCT1, which acts as a proton harvesting antenna facilitating monocarboxylate/proton co-transport [17].

A recent study on the shedding of basigin by a transmembrane protease in lung squamous cell carcinoma hints at a direct modulation of the L-lactate transport [16]. Upregulation of the protease was found to promote cleavage of the extracellular portion of basigin resulting in increased tumorigenesis by enhanced L-lactate efflux via MCT4, i.e. the Warburg effect. In this study, we generated functional fusion constructs of basigin variants and MCT1 (Fig 1A and S1 Fig), and show an intracellular accumulation of L-lactate caused by the membrane-proximal Ig-I domain. A correctly folded basigin Ig-I domain shifted the transmembrane distribution ratio of L-lactate by a factor greater than four at a given proton transmembrane gradient. We postulate that basigin uses the extracellular domain to create a microenvironment of increased proton and substrate concentrations at the transporter entry side allowing intracellular substrate accumulation according to Le Chatelier's principle. This generates an apparent shift of equilibrium with respect to buffer concentrations. In this study, the accumulation promoted L-lactate-dependent cell growth. This attributes a regulation mechanism to the basigin extracellular domain by directly affecting MCT-dependent L-lactate transport with decisive implications on tumor energy metabolism and progression.

## Results

### Fusion of basigin with MCT1 maintains the specific mode of interaction

Basigin homologs are naturally absent in yeast [25]; yet, certain heterologously expressed mammalian MCTs reach the plasma membrane and exhibit transport functionality [26–28]. We employed a yeast knockout strain that is devoid of endogenous monocarboxylate transporters (W303-1A jen1Δ ady2Δ) [29] as an MCT expression system (Fig 1B). Integration of

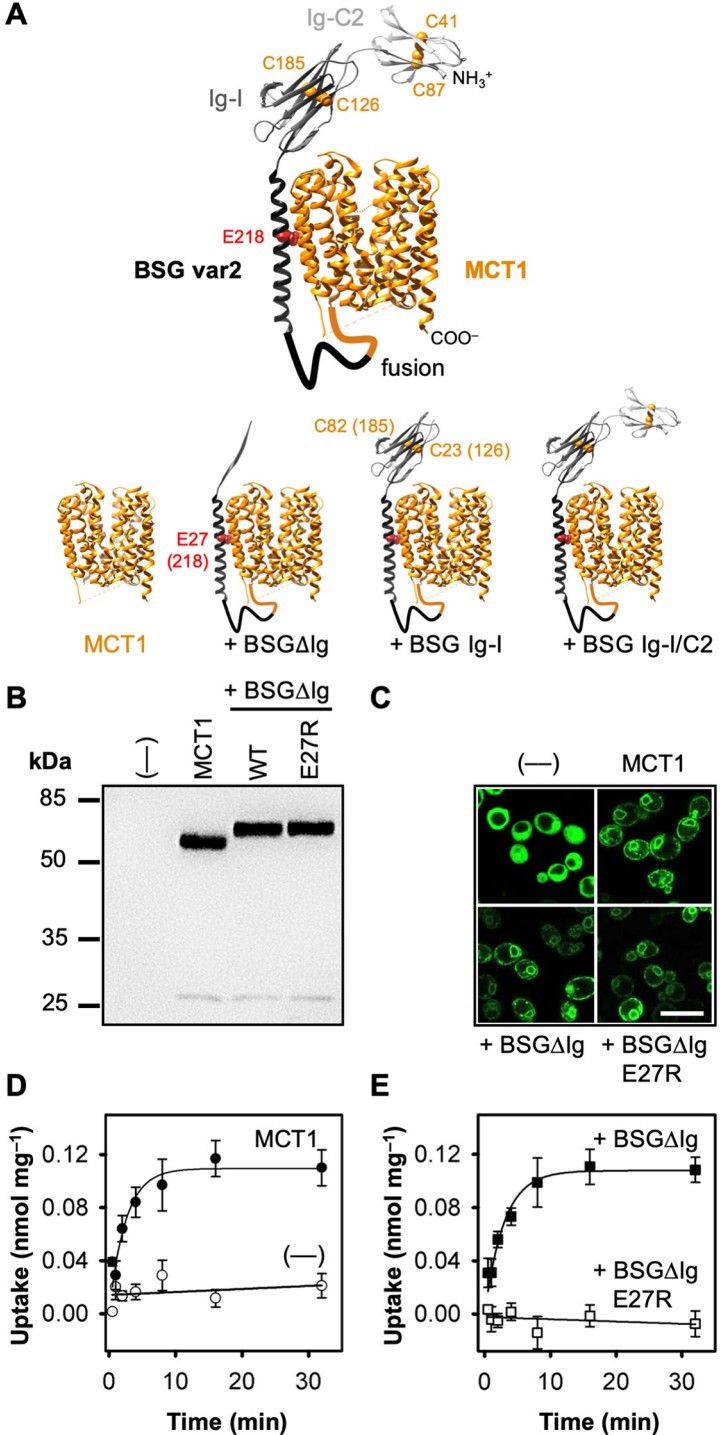

**Fig 1. Basigin-MCT1 fusion constructs, expression in yeast, and transport functionality.** (**A**) Shown are fusion constructs of basigin (black/gray; PDB #6LZ0) with MCT1 (orange; PDB #6LZ0). Disulfide bridges are shown as orange spheres, and Glu218 of the basigin transmembrane helix is highlighted in red. (**B**) Western blot showing expression of MCT1 alone (58 kDa), and fused with BSGΔIg (WT, 67 kDa) or the E27R point mutant. Proteins were detected using a Penta-His antibody. (**C**) Live-cell confocal microscopy of fusion proteins via C-terminal GFP and soluble GFP (—). (**D, E**) Uptake of $^{14}$C-labeled L-lactate into jen1Δ ady2Δ yeast at pH 6.8 and a 1 mM inward gradient. Shown are curves for non-expressing cells (background, D, ○), cells expressing MCT1 alone (D, ●), and in fusion with BSGΔIg (E, ■), or BSGΔIg E27R (E, □). The data are normalized to 1 mg of cells and the background of non-expressing cells was subtracted. Error bars indicate ± S.E.M. from three biological replicates.

**Table 1. Kinetic parameters ± S.E.M. of MCT1/basigin fusion variants (nD: Not determined).**

| Construct | Uptake rate* nmol mg$^{-1}$ min$^{-1}$ | Uptake capacity† nmol mg$^{-1}$ & (mM) | $K_m$ (mM) | $v_{max}$ (nmol min$^{-1}$ mg$^{-1}$) | Membr. loc. rel. intensity |
|---|---|---|---|---|---|
| MCT1 | 0.049 ± 0.005 | 0.11 ± 0.01 (0.39 ± 0.05) | 2.8 ± 0.2 | 0.187 ± 0.009 | 1.00 ± 0.13 |
| + BSGΔIg | 0.037 ± 0.004 | 0.11 ± 0.01 (0.39 ± 0.03) | 2.4 ± 0.7 | 0.125 ± 0.037 | 0.59 ± 0.07 |
| + BSG Ig-I | 0.105 ± 0.008 | 0.50 ± 0.01 (1.77 ± 0.04) | 3.9 ± 0.1 | 0.419 ± 0.024 | 1.66 ± 0.11 |
| + BSG Ig-I C23S | 0.012 ± 0.001 | 0.12 ± 0.02 (0.44 ± 0.07) | nD | nD | nD |
| + BSG Ig-I C82S | 0.007 ± 0.002 | 0.05 ± 0.01 (0.18 ± 0.05) | nD | nD | nD |
| + BSG Ig-I/C2 | 0.098 ± 0.013 | 0.59 ± 0.01 (2.11 ± 0.02) | 6.8 ± 3.1 | 0.431 ± 0.175 | 0.79 ± 0.06 |
| + BSG var2 | 0.021 ± 0.002 | 0.51 ± 0.06 (1.82 ± 0.20) | 1.2 ± 0.2 | 0.041 ± 0.006 | 0.58 ± 0.07 |
| + BSG Glu→Gln (114,118,120,168,172) | 0.069 ± 0.006 | 0.25 ± 0.01 (0.89 ± 0.04) | nD | nD | nD |
| + BSG Lys/Arg→Ala (108,111,127/201,203) | 0.005 ± 0.001 | 0.11 ± 0.01 (0.39 ± 0.03) | nD | nD | nD |
| + BSG Lys108,111Ala | 0.133 ± 0.031 | 0.47 ± 0.07 (1.70 ± 0.24) | nD | nD | nD |
| + BSG Arg201,203Ala | 0.013 ± 0.002 | 0.25 ± 0.01 (0.91 ± 0.01) | nD | nD | nD |

* dependent on the number of active transporters in the plasma membrane (1 mM external L-lactate, pH 6.8).

† independent of the number of active transporters (1 mM external L-lactate, pH 6.8).

the constructs into the plasma membrane was visualized via GFP-fusions (Fig 1C). The fluorescence signal of the circular structures in the cytosol represents MCT in the perinuclear region or yeast vacuole membrane. This unused protein portion does not affect transport events at the plasma membrane. For assaying transport of radiolabeled L-lactate, constructs without GFP were used. Contrarily to previously used mammalian cells and *Xenopus laevis* oocytes, yeast background transport and metabolism of L-lactate are virtually absent even after prolonged assay times due to the jen1Δ ady2Δ deletions and repression of L-lactate-converting cytochrome c oxidoreductase by glucose [30] (Fig 1D and S2 Fig). This allowed us to monitor L-lactate uptake until the equilibrium state was reached. This state is indicated by the plateau in the transport curves at which import and export velocities are equal. The plateau was typically observed within the experiments, or could be extrapolated from the data. The resulting equilibrium concentrations inside and outside of the cells are independent from the number of transporters present at the plasma membrane permitting direct comparison of different expression constructs irrespective of their individual expression levels [31,32].

Expression of MCT1 alone facilitated uptake at an initial rate of 0.05 nmol L-lactate per min and mg of yeast cells yielding a total cellular L-lactate load of 0.11 nmol mg$^{-1}$ in the steady state at minimal background (Fig 1D, Table 1). The MCT and basigin protein topologies seemed particularly suited for fusion as the linking sequence stretch amounts to about 60 amino acids providing for high flexibility and minimal constraint (S1 Fig). Protein fusion ensures an even stoichiometry and close proximity of both partners at all times. First, we fused an artificial minimalistic basigin variant to MCT1 harboring only the transmembrane span and the intracellular C-terminal domain (BSGΔIg; Fig 1A). Expression and localization of the BSGΔIg-MCT1 fusion protein was similar to that of unfused MCT1 (Fig 1B and 1C). Notably, L-lactate transport remained unaltered (Fig 1E, Table 1). We verified that the transmembrane domains of basigin and MCT1 were interacting by the typical binding mode by implementing a BSGΔIg mutation replacing Glu27 by Arg (BSGΔIg E27R; the position corresponds to Glu218 in BSG var2; Fig 1A). This glutamate interacts with Asn187 in transmembrane span 6 of MCT1 [12] and respective basigin mutants were shown before to fail to interact with MCT1 [33]. The BSGΔIg E27R-MCT1 fusion protein was well expressed in yeast and reached the plasma membrane (Fig 1B and 1C). However, confirmatively, L-lactate transport of the fused

MCT1 was abolished (Fig 1E). We assume that disruption of the interaction between basigin and MCT1 in the membrane domain at a sustained close link of the fused proteins interferes with a proper arrangement of the transmembrane spans leading to a loss of transport functionality. MCT1 transport functionality could not be rescued by adding the extracellular Ig-like domains to the mutated basigin demonstrating the importance of proper interaction in the transmembrane domain region (S3 Fig).

## The membrane-proximal Ig-I domain of basigin increases intracellular L-lactate concentrations

Sequential addition of the membrane-proximal Ig-I domain (BSG Ig-I), the Ig-C2 domain (BSG Ig-I/C2), and an N-terminal signal peptide (BSG var2) to the fusion construct (Fig 1A) generally yielded lower expression levels (Fig 2A), yet maintained plasma membrane localization (Fig 2B, Table 1). Heterologously expressed membrane proteins often remain unglycosylated in the yeast [25–28,31], and the obtained bands at correct sizes in the Western blot were indicative of nonmodified protein. Unexpectedly, the constructs greatly increased the L-lactate load of the cells in the transport equilibrium state despite unaltered substrate/proton concentrations in the external buffer. Expression of the BSG Ig-I construct manifested in a transport capacity of 0.50 nmol mg$^{-1}$, i.e. 4.5-fold higher than MCT1 alone or when fused with the BSGΔIg variant (Fig 2C; note the different scale of the y-axis when comparing to Fig 1D and 1E; Table 1). Further addition of the Ig-C2 domain in the BSG Ig-I/C2 and BSG var2 constructs resulted in equally high uptake capacities (Fig 2D). The observed differences in transport rates resulted from varying transporter numbers at the cellular surface (Fig 2B, Table 1). Due to lower expression levels the capacity of BSG var2 was extrapolated from the initial 32 min of uptake. Our data show that the extracellular Ig-I domain of basigin positively disturbs the inward gradient conditions generated by the provided external buffer possibly by increasing the local L-lactate and/or proton concentration at the MCT entry site.

Ig-domains consist of two β-sheets connected by a disulfide bridge (orange spheres in Fig 1A). We disrupted the disulfide bridge of the Ig-I domain by serine mutations (BSG Ig-I C23S and BSG Ig-I C82S; corresponding to positions 126 and 185 in BSG var2). The resulting fusion constructs with MCT1 remained functional (Fig 2E). However, the transport capacities were as low as with the constructs that lack the basigin Ig-I domain altogether (compare to Fig 1D and 1E, Table 1). The Western blot indicates some protein fragmentation of the BSG Ig-I C23S construct and weakly of BSG Ig-I C82S (Fig 2F). Yet, even a putatively mixed protein population should at least partially exhibit lactate accumulation properties if the BSG Ig-I domain is folded correctly. Since this was not the case, we conclude that the Cys mutations indeed rendered the Ig-I domain misfolded, leading to a loss of a substrate/proton-accumulating function.

## BSG Ig-I facilitates the use of protons and affects Michaelis-Menten kinetics

Yeast cells abundantly express a plasma membrane proton ATPase to regulate the cytosolic pH and to prevent acidification [34]. The intracellular pH of a yeast cell is typically mildly acidic, and external assay buffers are respectively adjusted to pH 6.8 to match the cytosolic pH conditions. Using a fluorescent pH indicator [31], we found that upon addition of 1 mM L-lactate the cytosolic pH of yeast lacking monocarboxylate transporters (Fig 3A), or expressing MCT1 (Fig 3B) or the BSG Ig-I fusion construct (Fig 3C) stabilized around pH 6.3–6.5 in all cases. The measurements confirm that efficient pH regulation mechanisms are in place in yeast cells. At our standard assay conditions (external pH of 6.8), the proton transmembrane gradient in

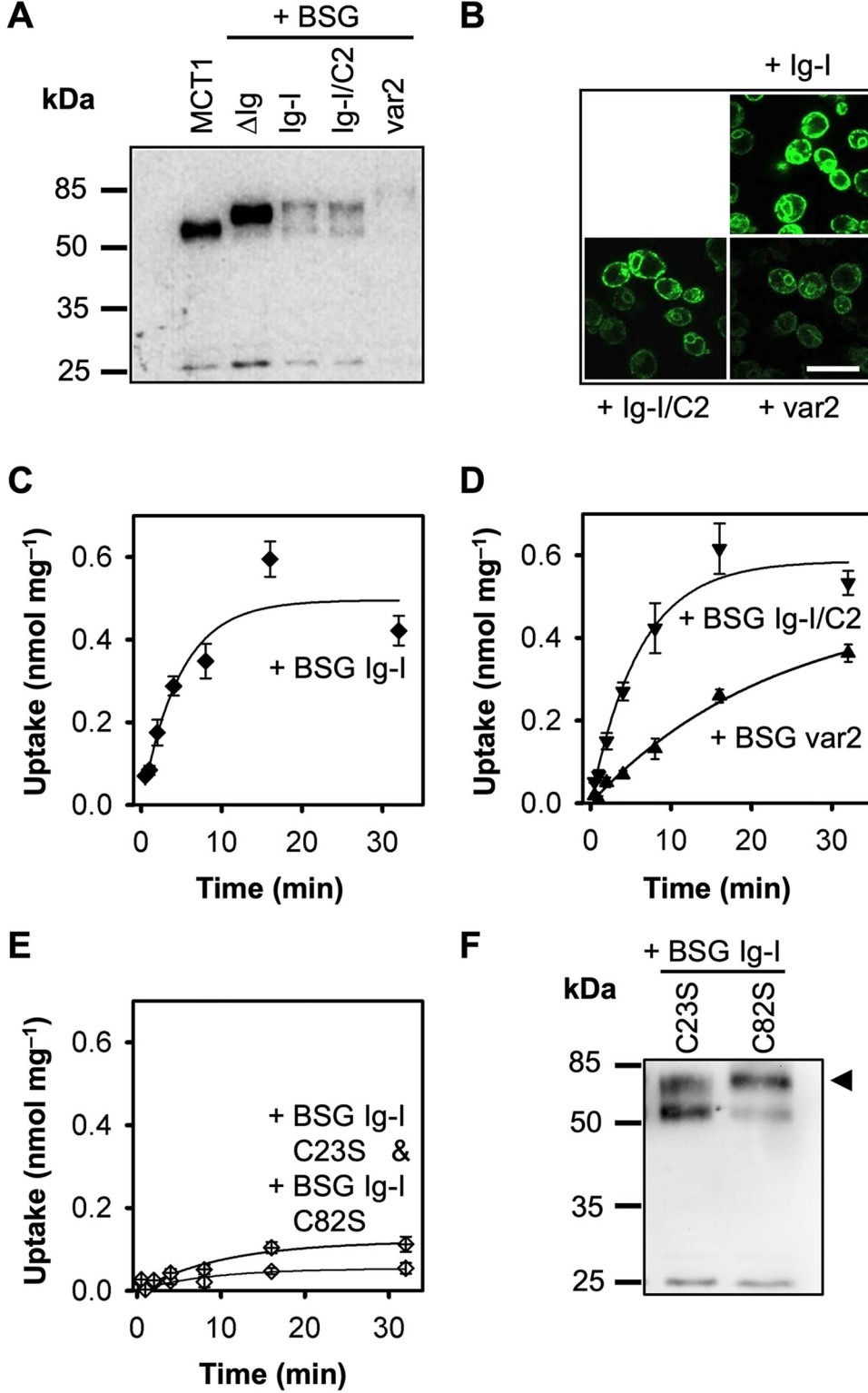

**Fig 2. Increased uptake of L-lactate by MCT1 fusions with basigin variants carrying the extracellular Ig-I domain.**
(**A**) Western blot showing expression of MCT1 alone (58 kDa) and fused with BSGΔIg (67 kDa), BSG Ig-I (77 kDa),
BSG Ig-I/C2 (85 kDa), and BSG var2 (87 kDa) using a Penta-His antibody. (**B**) Visualization of the constructs at the
plasma membrane via C-terminal GFP and live-cell confocal microscopy. (**C–E**) Uptake of $^{14}$C-labeled L-lactate via
MCT1 fusion constructs with basigin variants at pH 6.8 and a 1 mM inward gradient. Shown are BSG Ig-I (C, ◆), BSG

Ig-I/C2 (D, ▼), BSG var2 (D, ▲), and the BSG Ig-I point mutants C23S (E, ◔) and C82S (E, ◇). The background of non-expressing cells was subtracted, and the data were normalized to 1 mg of cells; error bars represent ± S.E.M. of three biological replicates. (**F**) Western blot showing expression of MCT1 fused to BSG Ig-I C23S and C82S. The arrow head indicates the expected molecular weight of 77 kDa (Penta-His antibody).

the equilibrium state is, thus, directed outward yet very flat ($\approx 0.16$ μM [H$^+$] external vs. $\approx 0.4$ μM [H$^+$] internal, i.e. a steepness of $\approx 2.5$).

According to Le Chatelier's principle, an additional provision of protons as a co-substrate to the cis-side (pH 5.8) generated a steeper inward transmembrane pH gradient ($\approx 1.6$ μM [H$^+$] external) and increased the total intracellular L-lactate load (Fig 3D). Again, fusion with BSG Ig-I resulted in considerably higher cellular concentrations of L-lactate compared to MCT1 alone (Fig 3D). The relatively stronger effect at pH 6.8 (4.5-fold) versus pH 5.8 (2-fold) indicates that the influence of the BSG Ig-I domain on L-lactate/H$^+$ co-transport is higher when protons are scarce (Fig 3E).

To better grasp the nature of the MCT1 transport modulation by basigin, we determined Michaelis-Menten kinetics (Fig 3F). Resulting $K_m$ and maximum transport velocity values

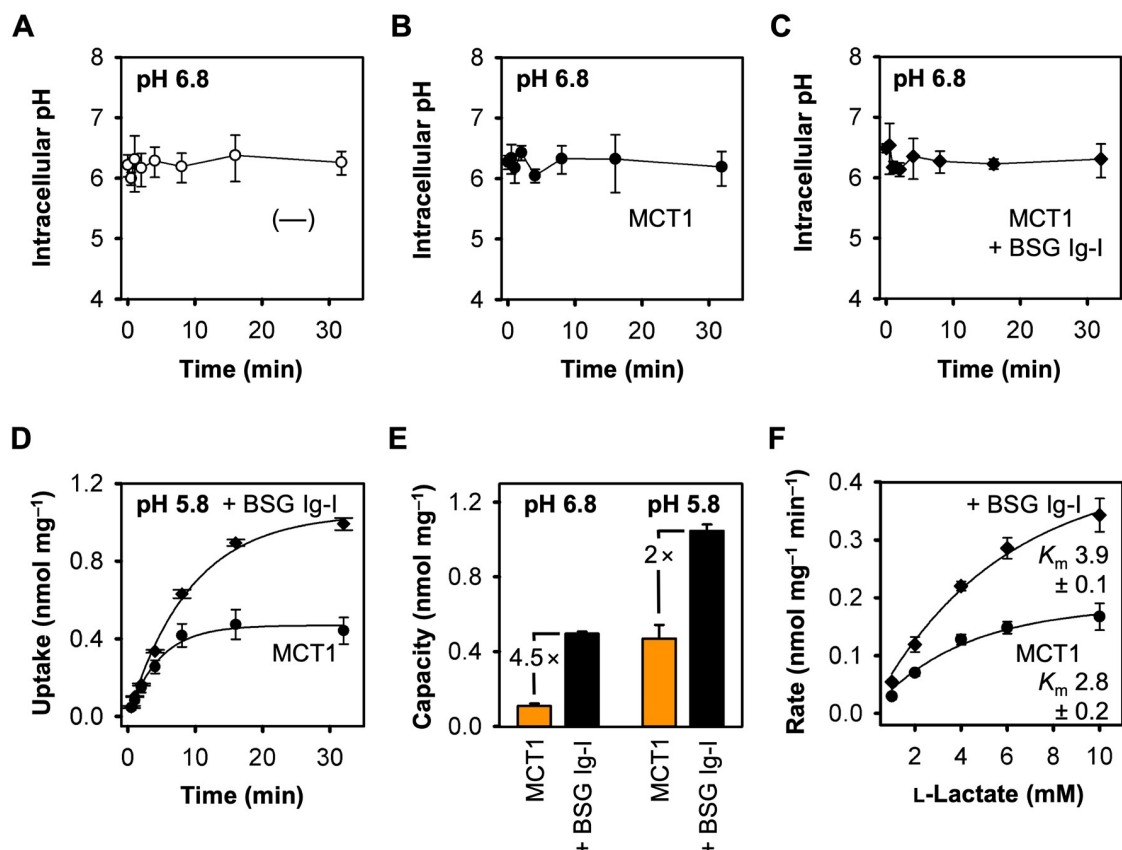

**Fig 3. Effect of the transmembrane proton gradient and altered substrate concentrations on transport via MCT1 in the presence and absence of the basigin Ig-I domain.** (**A-C**) Effect of adding 1 mM L-lactate at an external buffer pH of 6.8 to yeast cells without monocarboxylate transporters (A, ○), or expressing MCT1 (B, ●) or MCT1 fused with BSG Ig-I (C, ◆). (**D**) Uptake of $^{14}$C-labeled L-lactate via MCT1 alone (●) and fused wit BSG Ig-I (◆) at an external pH 5.8 and a 1 mM inward L-lactate gradient. (**E**) Uptake capacities of cells expressing MCT1 alone (orange) and fused with BSG Ig-I (black) at pH 6.8 (based on data from Figs 1D and 2C) and pH 5.8 (data from 3D). (**F**) Michaelis-Menten kinetics for MCT1 alone and fused with BSG Ig-I. $K_m$ values were determined at pH 6.8 and are given in mM. In all cases, the background of non-expressing cells was subtracted; error bars indicate ± S.E.M. of three biological replicates.

($v_{max}$) are summarized in Table 1 (S4 Fig for remaining constructs). For MCT1 alone, we determined a $K_m$ of 2.8 ± 0.2 mM. The $K_m$ significantly increased to 3.9 ± 0.1 mM (p = 0.01, unpaired, two-tailed Student's t-test) in the presence of BSG Ig-I indicating lower apparent substrate affinity or a modulation of the substrate availability that differs from the overall buffer concentration (Fig 3F). To determine $v_{max}$, we estimated the amount of transporter at the plasma membrane from the intensity signal of respective constructs carrying a C-terminal GFP. Here, fusion with BSG variants containing the Ig-I domain appeared to increase $v_{max}$ (Table 1).

## Basigin-driven L-lactate uptake exceeds the external buffer substrate concentration

Determination of the molar L-lactate concentration ratio of the intracellular space and the external buffer, $c_{eq\_in}/c_{eq\_ex}$, required knowledge of the substrate-accessible yeast volume. To gain this information, we loaded cells with $^{14}$C-labeled glycerol via heterologously expressed aquaglyceroporins, AQP7 and AQP9, ensuring equal internal and external concentrations at equilibrium, $c_{eq\_in} = c_{eq\_ex}$, because diffusion of neutral solutes through aquaglyceroporin channels is solely gradient-driven, non-directional, and equilibrative [35]. From two different buffer glycerol concentrations, 1 mM and 5 mM, we derived the conversion relation of 1 mM intracellular substrate corresponding to 0.28 ± 0.03 nmol mg$^{-1}$ of cells (Fig 4A). At $8.6 \times 10^6$ cells per mg [31], this gave a yeast cell volume of 33 fl matching previous measurements that range from 31–46 fl [36,37]. For the molar intracellular and extracellular L-lactate concentration ratios ($c_{eq\_in}/c_{eq\_out}$) resulting from conversion of this work's transport data see Fig 4B and Table 1. MCT1 alone and the constructs that are devoid of a correctly folded basigin Ig-I domain facilitated L-lactate transport to less than half of the external buffer concentration (0.39–0.44 mM; the value for the weakly active BSG Ig-I C82S construct falls somewhat short at 0.18 mM). Taking into account the previously determined slightly outward-directed transmembrane proton gradient ($\approx$ 2.5: 1), this shows that at around 0.4 mM cytosolic L-lactate the system was equilibrated ($c_{eq\_in} \cdot [H^+]_{eq\_in} = c_{eq\_ex} \cdot [H^+]_{eq\_ex}$ with $K = 1$; Fig 4C). Oppositely, the fusion constructs containing a functional Ig-I domain led to an accumulation of intracellular L-lactate (1.8–2.1 mM) above the concentration of 1 mM provided with the buffer. Here, the established L-lactate concentration ratio (cytosol vs. buffer) combined with the transmembrane proton gradient indicates a disturbance of the system by a factor of 4.5 ($K_{apparent}$ = 4.5; Fig 4C). Thermodynamically, such a disturbance would derive from a local increase of the substrate/proton concentration at the extracellular side of the basigin/MCT1 complex above the bulk buffer concentration but without affecting the kinetic properties of MCT1 itself.

Next, we asked whether the observed degree of L-lactate accumulation in the cytosol is physiologically meaningful by monitoring L-lactate-dependent cell growth (Fig 4D). In fact, expression of MCT1 alone or in fusion with basigin lacking a functional Ig-I domain hardly facilitated growth when L-lactate was provided externally as the sole energy and carbon source. However, growth of cells expressing constructs with a properly folded Ig-I domain (BSG Ig-I, BSG Ig-I/C2, BSG var2) was strongly promoted under these conditions. The phenotypic data, thus, perfectly mirror the biophysical assays and confirm an accumulating effect of the basigin Ig-I domain on the intracellular L-lactate content.

## The basigin Ig-I domain acts as a proton and L-lactate harvesting antenna

To gain insight into the mechanism by which basigin leads to an intracellular L-lactate accumulation we investigated the electrostatics of the Ig-I domain. For this purpose, the recent cryo-EM structure of basigin in complex with MCT1 [12] was not usable due to low resolution

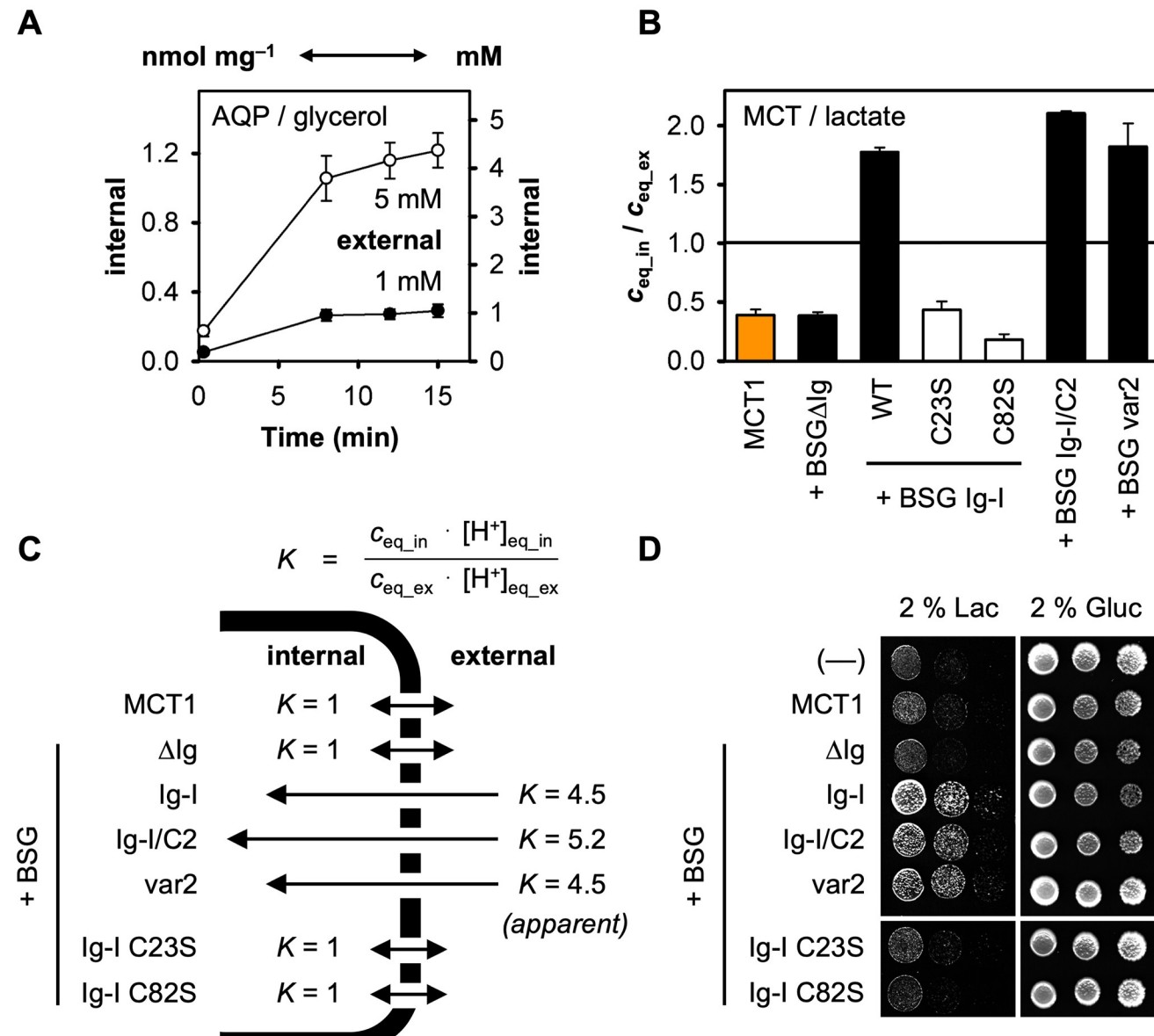

**Fig 4. Shifted transmembrane L-lactate distribution and promotion of cell growth by the presence of the BSG Ig-I domain.** (**A**) Conversion of scales from the amount of intracellular substrate per cell mass to the molar concentration (nmol mg$^{-1}$ to mM) by use of the freely diffusible aquaporin substrate glycerol. Uptake of $^{14}$C-labeled glycerol of AQP7 or AQP9 expressing cells was monitored for 15 min at pH 6.8 and 1 mM (●) and 5 mM (○) inward gradients. The error bars denote ± S.E.M. from four biological replicates. (**B**) Intracellular L-lactate accumulation generated by expression of MCT1 alone (orange) and fusions with basigin variants (black); error bars denote ± S.E.M. from three biological replicates. (**C**) Relation of the internal/external L-lactate ($c_{eq\_in}$, $c_{eq\_ex}$) and proton concentrations ([H$^+$]$_{eq\_in}$, [H$^+$]$_{eq\_ex}$) at equilibrium. The tested basigin-MCT1 fusion constructs yielded different ratios (at 1 mM L-lactate, pH 6.8 conditions) indicating an equilibrative ($K = 1$) or accumulating effect (apparent $K > 1$) on intracellular L-lactate with respect to the overall buffer concentrations. (**D**) Growth of cells expressing MCT1 and basigin fusion variants on agar media containing 2% sodium L-lactate (left) or 2% glucose (growth control; right). Cell suspensions (5 µl) were spotted in ten-fold serial dilutions from a starting OD$_{600}$ of 1.

in the area of the relevant charged residues of which sidechains are missing in the structure data. We therefore calculated the Poisson-Boltzmann electrostatic potential of the Ig-like domains using high-resolution crystal data (PDB #3B5H; Fig 5A). This revealed a strongly negative patch of packed glutamate residues close to the membrane, which meets the requirements of a proton harvesting antenna. Quite unexpectedly, an oppositely charged patch is situated

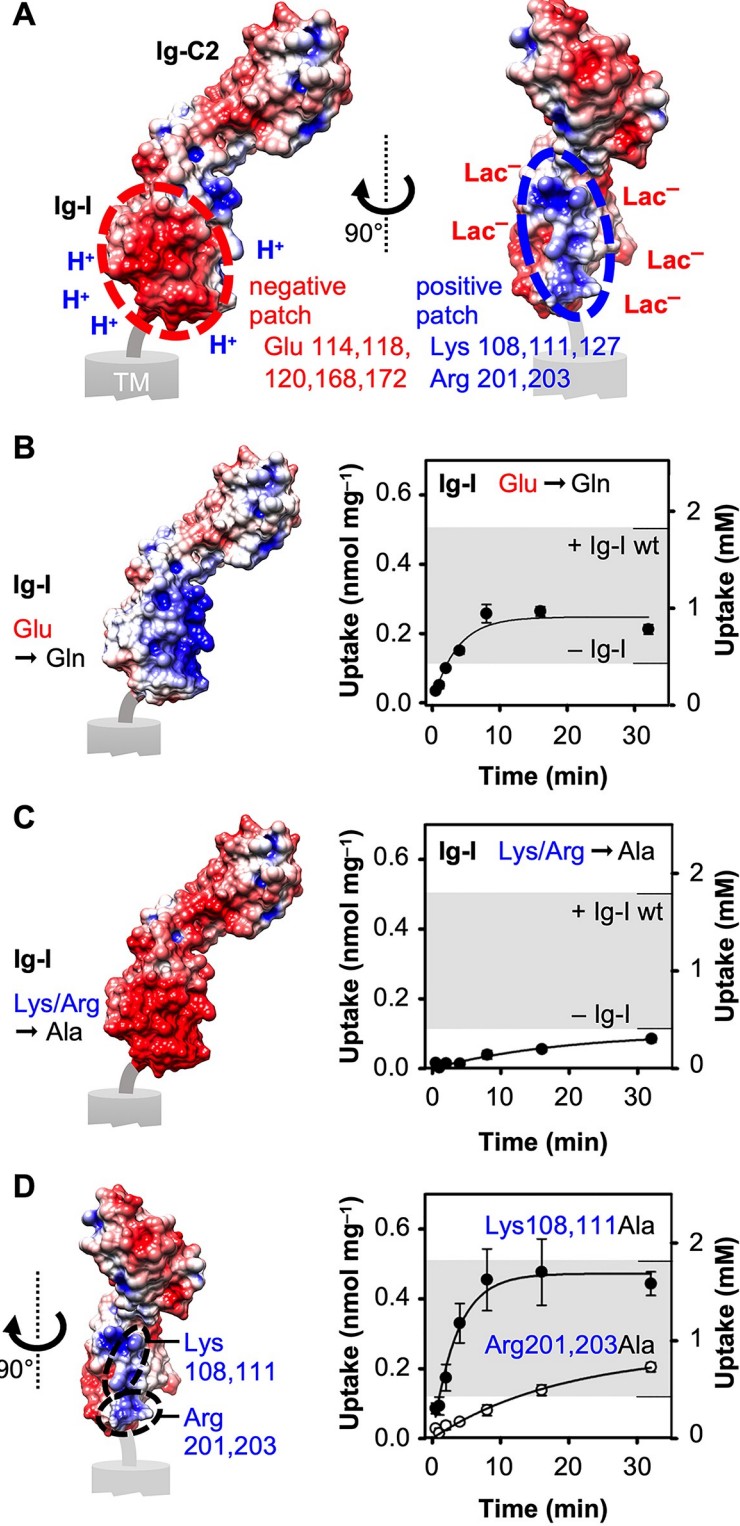

**Fig 5. Two oppositely charged surface patches in the BSG Ig-I domain.** (**A**) Poisson-Boltzmann electrostatic potential of the extracellular domain of BSG var2 (PDB #3B5H) reveals two oppositely charged surface patches in the BSG Ig-I domain; the red/blue scale covers the range of −3 kT e⁻ to +3 kT e⁻. (**B–D**) Effect of replacing charged residues in the BSG Ig-I domain by neutral ones on the electrostatic potential (left) and the L-lactate transport of fused MCT1 (right). The five Glu residues of the negative patch (pos. 114, 118, 120, 168, 172) were changed to Gln (B, ●),

the five Lys/Arg residues of the positive patch (pos. 108, 111, 127, 201, 203) were mutated to Ala altogether (C, ●) or in two pairs of two (D, Lys108,111Ala: ●, Arg201,203Ala: ○). The electrostatic surface in D shows wild-type BSG for better orientation. The gray shading indicates the corridor between the uptake capacities of MCT1 fused with wild-type BSG Ig-I (upper border, from Fig 2C) and MCT1 alone (lower border, from Fig 1D).

right next to this area that is made up of lysine and arginine residues and might attract L-lactate anions (Fig 5A). We generated basigin mutants in which either the five clustered glutamates of the negative patch (Glu114,118,120,168,172) were changed to neutral glutamine residues, or the five lysine and arginine residues of the positive patch (Lys108,111,127; Arg201,203) were replaced by alanine (S1 and S5 Figs). The respective electrostatic surface potentials are visualized in Fig 5B and 5C. Both basigin mutants when fused with MCT1 resulted in functional transport proteins which we used to determine the L-lactate uptake capacity. Removal of the negative charges in the Ig-I domain led to less accumulation of intracellular L-lactate compared to the wild-type basigin Ig-I domain (Fig 5B, upper border of the indicated corridor) corroborating a proton harvesting role of the negative patch. However, the achieved internal L-lactate concentration of 0.89 mM was still higher than with MCT1 alone (Fig 5B, lower border of the indicated corridor). This is indicative of an additional harvesting function of the Ig-I domain, possibly for L-lactate anions by the positive patch. Indeed, replacement of all five positive residues by neutral alanines greatly decreased the L-lactate uptake capacity (Fig 5C). The resulting load of intracellular L-lactate was extrapolated from the initial 32 min of the uptake curve. This gave an equal level as with MCT1 alone (Table 1) indicating a full loss of substrate accumulation. We generated two more basigin mutant constructs in which only two positive residues were replaced at a time by neutral alanine (Ig-I Lys108,111Ala and Ig-I Arg201,203Ala; Fig 5D). These mutants showed that Arg201 and Arg203 that are more proximal to the membrane and, thus, closer to the MCT1 transporter entry site are particularly relevant for promoting intracellular L-lactate accumulation (Fig 5D). While mutation of the two lysines decreased the uptake capacity by less than 10%, exchange of both arginines diminished equilibrium L-lactate uptake as strongly as the basigin construct lacking all five negative charges of the glutamates (Table 1). Together, the Ig-I domain locally increases the concentration of protons and more prominently of L-lactate anions mainly by a membrane-proximal double-lysine motif.

## Discussion

Earlier studies demonstrated that trafficking of MCTs to the plasma membrane critically depends on complexation with basigin or embigin [8,14,38,39]. Altered MCT membrane localization was shown as causative in disease contexts, mainly cancer [8] and recently Diabetes mellitus [39], due to altered L-lactate transport capabilities of the cell. Basigin executes the same trafficking function on a plasma membrane $Ca^{2+}$-ATPase (PMCA); as in the basigin-MCT complex, interaction in the transmembrane regions seems sufficient for correct trafficking [40,41].

Investigations of a putative direct modulation of the MCT transport properties by basigin using *X. laevis* oocytes or mammalian cells suffer from interference with endogenous MCTs and basigin. In these systems, rapid metabolism of L-lactate restricts measurements to the seconds time range. Likewise, determination of lactate-coupled proton facilitation will yield a transient view on the transport situation due to rapid buffering and $H^+$-ATPase action. The use of yeast with virtually absent metabolic conversion of L-lactate, however, allowed us to monitor radiolabeled L-lactate transport at physiological concentrations and for prolonged times that were mostly sufficient to reach the equilibrium state regarding the transmembrane L-lactate distribution. The absence of endogenous L-lactate transporters gave minimal

background. The equilibrium state is characterized by a certain defined intracellular/extracellular substrate/co-substrate concentration ratio independent of the number of transporters residing at the plasma membrane. In connection with the fact that the transmembrane domains of basigin and MCT1 fall into proper arrangement when expressed as fusion proteins, our setup permitted safe conclusions about the direct modulation of cellular L-lactate uptake by basigin even at the domain level. This disclosed an overlooked basic function of the membrane-proximal Ig-I domain of basigin in facilitating an intracellular accumulation of L-lactate above the provided external substrate/proton concentrations.

Quite recently, with carbonic anhydrase (CA) IV, a new interacting partner of the basigin Ig-C2 domain was identified [17]; a second CA isoform, CAII, was found to bind to the intracellular C-terminus of MCT1 [42,43]. In this context, first experimental evidence was obtained for the indirect modulation of MCT transport by regulation of the local proton availability as both CAs appear to act as proton harvesting antennae. An unexplained mismatch between a mathematical model on proton-activated MCT transport and experimental data led to the conclusion that MCT1 itself may have proton-collecting properties or additional partner proteins may be involved [44]. Recent findings indicate a role of aspartate residues at the extracellular faces of MCT1, MCT2, and MCT4 in the recognition of protons and substrate [45].

Our calculation of the Poisson-Boltzmann electrostatic potential of basigin indicated that the Ig-I domain may in fact display harvesting and local concentration properties for protons (Fig 5A). Experimental replacement of charged residues by neutral ones corroborated this notion (Fig 5B–5D). At the same time and even more prominently, the Ig-I domain harvests monocarboxylate substrate anions at the extracellular side of the plasma membrane. Previously, we found a related electrostatic attraction mechanism for L-lactate anions to locally accumulate at a positive arginine cluster at the surface of the human AQP9 tetramer increasing uptake of concomitantly elevated levels of neutral lactic acid via AQP9 [46]. In basigin, unprecedentedly, a peculiar side-by-side placement of negative and positive surface patches attracts both, the L-lactate substrate and the proton co-substrate, to the MCT1 transporter surface generating a locally increased availability and a steeper inward gradient than the external bulk buffer (Fig 6). The loss of the bivalent antenna function by disulfide bridge-breaking point mutations is likely due to dissolution of the spatial organization of the negative and positive residue sidechains.

Recent findings in a physiological setting are perfectly in line with our data on shifted L-lactate accumulation caused by the BSG Ig-I domain [16]. Shedding of the extracellular domain of basigin by action of the transmembrane protease TMPRSS11B altered the equilibrium L-lactate efflux via MCT4 by a factor up to 3.5; in our study, the transmembrane L-lactate distribution involving MCT1 was shifted 4.5-fold (Figs 3E and 4C). The similar effect on two different MCT isoforms is noteworthy and suggests that the basigin Ig-I domain by its mere proximity to the MCT creates a microenvironment of locally increased extracellular concentrations of protons and L-lactate (Fig 6). Like the proton harvesting function by CA IV being even farther away from the MCT entry site [17], the large spacing of the charged residues of basigin in the 6–10 Å range renders a residue-specific type of direct substrate shuttling or proton-wiring into MCT1 or MCT4 unlikely. Charged residues at the surface of the adjacent MCT may, however, contribute to the accumulation effect [11,45]. In fact, a cluster in MCT1 carrying three negatively charged glutamates (Glu46, Glu48, Glu57) and two positively charged residues (Lys45, Arg175) is located at the interface with the basigin Ig-I domain (Arg201, Arg203; see S6 Fig). We conclude that a local increase in the availability of protons and L-lactate constitutes a disturbance according to Le Chatelier's principle in which the generally unaltered equilibrium state reaches higher intracellular concentrations. In the microenvironment, thermodynamic

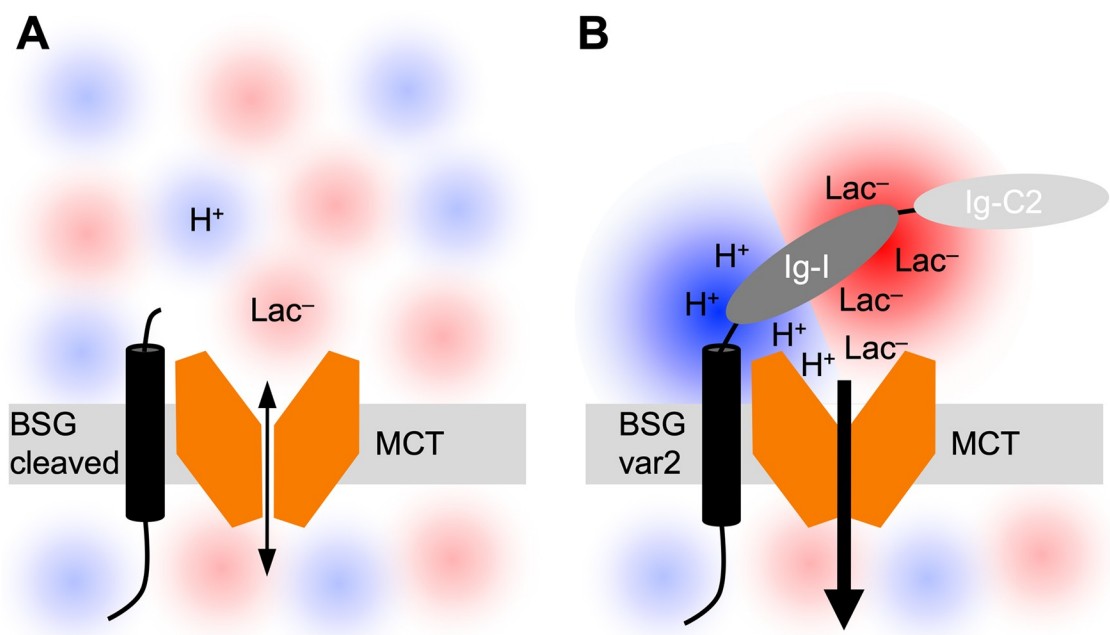

**Fig 6. Bivalent harvesting function of the BSG Ig-I domain for protons and ʟ-lactate anions.** (**A**) In the absence of the extracellular BSG Ig-I domain, the environmental concentrations of ʟ-lactate and protons determine transport. (**B**) The presence of the Ig-I domain creates a microenvironment and increases the local concentration of the substrate and protons at the MCT entry site driving accumulation in the cell.

requirements $K = (c_{eq\_in} \cdot [H^+]_{eq\_in})/(c_{eq\_ex} \cdot [H^+]_{eq\_ex})$ are met, yet enabling an shifted apparent equilibrium state with regard to the overall "macroenvironment".

The ubiquitous intermediate-affinity MCT1 is promiscuous in that it assumes roles in ʟ-lactate-releasing as well as ʟ-lactate-depending cancer cells [47,48]. Here particularly, an altered transmembrane distribution of ʟ-lactate will be decisive. In a broad perspective, action of the basigin Ig-I domain would be generally prognostic for tumor progression that depends on lactate-transporting MCT isoforms, and pose a potential target for therapeutic intervention.

## Material and methods

### Expression plasmids, cloning and mutations

Human MCT1 (GenBank NM_001166496) and basigin variant 1 (GenBank NM_001728) were obtained as ORF clones in pcDNA3.1+/C-(K)DYK (GenScript). The yeast expression plasmid pDR196 (Addgene #36029) [49] was previously modified to encode an N-terminal hemagglutinin epitope tag and a C-terminal His$_{10}$ affinity tag. The coding sequences were introduced by PCR using primers that carry respective nucleotide extensions. The MCT1 ORF was cloned into pDR196 using PflMI and BspEI restriction sites introduced via a pair of oligonucleotides (S1 Table, "MCT1 linker"). The basigin variants BSGΔIg, BSG Ig-I, BSG Ig-I/C2 and BSG var2 were produced via PCR with additional SpeI and PstI restriction sites (S1 Table). The BSG Ig-I variants in which the glutamates at positions 114, 118, 120, 168, 172 were changed to glutamine, or the lysines 108, 111, 127 and arginines at 201 and 203 were replaced by alanine, were purchased as synthetic DNA fragments with additional SpeI and PstI restriction sites (GenScript). The basigin variants were inserted upstream of the coding region for MCT1 into SpeI and PstI sites. Site-directed point mutations were introduced using

oligonucleotides with respective nucleotide exchanges (S1 Table) according to the Quik-Change protocol (Agilent). All constructs were sequenced for verification. See S1 Fig for the exact protein sequences derived from all expression constructs. To obtain constructs with a C-terminal green fluorescent protein (yEGFP), the open reading frames were inserted into the SpeI and XhoI-compatible SalI sites of the pUG35 plasmid (U. Güldener and J. H. Hegemann, University of Düsseldorf, Germany). Expression constructs of human AQP9 and mouse AQP7 in pDR196 were generated previously [46].

## Yeast transformation and culture

*S. cerevisiae* W303-1A jen1Δ ady2Δ (MATa, can1-100, ade2-loc, his3-11-15, leu2-3,-112, trp1-1-1, ura3-1, jen1::kanMX4, ady2::hphMX4) yeast cells [29] lacking endogenous monocarboxylate transporters were kindly provided by M. Casal (Braga, Portugal). Cells were transformed using the lithium acetate/single stranded carrier DNA/polyethylenglycol method [50]. Transformed cells were selected and kept at 29˚C in selective SD medium supplemented with adenine, histidine, leucine and tryptophan, and 2% (w/v) glucose but lacking uracil. For assessment of cell growth on lactate agar plates, glucose was replaced by 2% L-lactate/$Na^+$ and buffered to pH 5.8 by 50 mM MES. The liquid lactate medium contained 1% L-lactate/$Na^+$. The number of yeast cells per $OD_{600}$ was counted in triplicate each at three densities (0.45, 0.50 and 0.55) using a Neubauer chamber with an Olympus IX50 microscope.

## Isolation of microsomal fractions, SDS-PAGE, and Western blot

40 ml yeast cultures were grown to an optical density ($OD_{600}$) of 1.0 ± 0.1, collected (4,000 g, 5 min, 4˚C) and washed with 50 ml of ice-cold water and 10 ml of ice cold TE buffer (25 mM Tris-HCl, 5 mM EDTA, pH 7.5); pellets were frozen and stored for at least 30 min at –80˚C. For disruption, the cell pellets were resuspended in 0.5 ml TE buffer and vortexed with 0.5 g of acid-washed glass beads (Ø 425–600 μm, Sigma-Aldrich) twelve times for 30 s each with 1 min on ice in between. Suspensions were cleared by centrifugation at 1,000 g for 5 min at 4˚C. The microsomal fractions were obtained by removal of high-density components at 10,000 g, 5 min, 4˚C, and ultracentrifugation at 100,000 g, 45 min, 4˚C. Membrane pellets were resuspended in 0.1 ml phosphate buffer (100 mM sodium phosphate, 50 mM NaCl, pH 8.0) for determination of the total protein content using the Bradford Protein Assay (Bio-Rad), and for separation by reducing SDS-PAGE. 5 μg total protein were loaded per lane. For Western blots, the proteins were blotted on PVDF membranes (Hybond P 0.45, GE Healthcare), and detected via a monoclonal mouse penta-His antibody (Cat. Nr. 34660, Qiagen) and a horseradish peroxidase-conjugated secondary goat-anti-mouse antibody (Cat. Nr. 115-035-174, Jackson ImmunoResearch) with the Clarity ECL substrate detection system (Bio-Rad). Visualization was done with a Lumi-Imager F1 (Roche) and a Chemostar Touch ECL & Fluorescence Imager (Intas Science Imaging Instruments).

## Live-cell confocal imaging

Yeast cells were collected at an $OD_{600}$ of 1.0 ± 0.1, washed and resuspended in water for imaging using a Leica SP5 confocal microscope with a HCX PL APO CS 63.0 × 1.40 Oil objective. The GFP-fluorophores were excited by an argon laser at 488 nm (14% LP power) and emission was monitored in the range of 500–540 nm. Images were taken in a window of 27.39 × 27.39 μm (512 × 512 px, 8 bit gray scale) with up to 8 confocal layers covering a vertical depth of 5.879 μm to 7.904 μm. The amount of expressed transport protein at the cell surface was determined semi-quantitatively using the ImageJ software (National Institutes of Health, USA) from confocal layers capturing cells in a central plane. The mean gray value per pixel

normalized to 1 μm vertical depth of individual cells was determined along an ellipsoid line running within the area of the plasma membrane. At least ten individual cells from five different areas were analyzed. Given errors denote S.E.M.

## Substrate transport assays

The uptake of L-lactate and glycerol was assayed as described before [51]. Briefly, yeast cells were harvested (4,000 g, 5 min, 4°C) at an $OD_{600}$ of 1.0 ± 0.1, washed once with ice cold water and resuspended in 50 mM HEPES/Tris, pH 6.8, or 50 mM MES/Tris, pH 5.8, as indicated, to a final $OD_{600}$ of 50 ± 5. For each measurement, 80 μl suspension containing 5.6 mg yeast cells were transferred into 1.5 ml tubes and brought to 19°C. Transport was initiated by the addition of 20 μl of substrate solution. To assay L-lactate uptake, a final concentration of 1 mM spiked with 0.04 μCi $^{14}$C- L-lactate (Hartmann Analytics) was used and uptake was monitored unshaken at 19°C for up to 32 min. For Michaelis-Menten kinetics, final L-lactate concentrations of up to 10 mM were used as indicated each containing 0.1 μCi $^{14}$C- L-lactate, and transport rates were determined at 2 min. For glycerol uptake, inward gradients of 1 mM and 5 mM supplemented with 0.04 μCi $^{14}$C-glycerol (Hartmann Analytics) were used and uptake was monitored for up to 15 min. Transport was stopped by abrupt addition of 1 ml of ice-cold water, rapid transfer of the sample to a GF/C glass microfiber filter (GE Healthcare), vacuum filtration, and washing with 7 ml of ice-cold water to remove excess substrate. Filters were analyzed in 3 ml of scintillation cocktail (Quicksafe A, Zinsser Analytic) with the Packard TriCarb 2900TR liquid scintillation counter (Perkin Elmer). Scintillation counting was done for 2 min. The amount of substrate that was taken up by the cells was calculated from: $amount_{sample}$ (nmol) = ($cpm_{sample}/cpm_{total}$) · $amount_{total}$ (nmol). Mean background counts of non-expressing cells were 104 cpm; the samples gave values from 200–579 cpm depending on the incubation time.

## Determination of the cytosolic pH in yeast

Yeast cells were collected at an $OD_{600nm}$ of 0.8 ± 0.05, washed in an equal volume of water, and resuspended in 50 mM HEPES/Tris, pH 6.8. The cells were subsequently incubated at 37°C for 24 h under agitation in an equal volume of 100 μM 5′(6′)-carboxyfluorescein-diacetatesuccinimidyl-ester (CFDA-SE, Sigma Aldrich) in 50 mM HEPES/Tris, pH 6.8, with 2% DMSO [52]. Extracellular CFDA-SE was removed by centrifugation at 10,000 g for 5 min, and cells were resuspended in 50 mM HEPES/Tris, pH 6.8, for a total of three times ensuring absence of excess or putatively leaked dye in the buffer [31]. Fluorescence was excited at $\lambda_{ex1}$ = 435 nm (isosbestic point) and $\lambda_{ex2}$ = 495 nm (pH-dependent), and emission intensity was determined at $\lambda_{em}$ = 525 nm (LS 55 fluorometer with a QS 4/4 mm quartz cuvette, Perkin Elmer) before and after addition of 1 mM L-lactate at room temperature. Each expression construct was measured in triplicate. The internal pH values were determined from the emission ratios by using the previously established calibration function $f(x) = -3.4238 + 5.4077\,x - 1.9197\,x^2 + 0.2033\,x^3$ [31].

## Structure models and electrostatic potential

Protein structure data for BSG2-MCT1 [12] (PDB #6LZ0) and the extracellular domains of basigin [18] (PDB #3B5H) were from the RCSB Protein Data Bank [53] and visualized with the PyMOL Molecular Graphics System (Schrödinger, LLC) and Chimera [54]. For Poisson-Boltzmann electrostatics, the PDB2PQR [55,56] and APBS tools [57] were used from within Chimera. Settings were: pH 7.0, protonation state assignment by PROPKA, and PARSE force-field. The topology plot was generated using TeXtopo [58].

## Statistical analysis

All curve fittings were done and visualized with SigmaPlot (Systat Software). Time-course curve fittings for L-lactate transport were done with the function $f(x) = a \cdot (1 - e^{-kt})$. Transport rates were determined from the slope at $t = 0$ (nmol min$^{-1}$ mg$^{-1}$). Maximal uptake capacities were calculated from the upper limit of each curve ($\lim_{t \to \infty} f(t) = a$). L-lactate uptake measurements were done in three biological replicates each with two or three technical repetitions. For glycerol uptake measurements four biological replicates were done each as an average from technical duplicates with AQP7 and AQP9. Michaelis-Menten constants ($K_m$), maximum velocities ($v_{max}$), transport rates and uptake capacities were calculated from individual single-exponential fits of each biological replicate and represent mean ± S.E.M. The amount of transporter present at the cell surface were derived from confocal GFP-intensity of the basigin fusion constructs relative to unfused MCT1 of which the mean gray value per pixel at 1 μm confocal depth was set to 1.

## Supporting information

**S1 Fig. Topology and protein sequence of the MCT1 fusion constructs with basigin variants.** Highlighted are the N-terminal hemagglutinin epitope tag (pink) and the C-terminal His10 affinity tag (blue), the cysteine residues forming disulfide bridges in the Ig-like domains Ig-I (dark grey) and Ig-C2 (light grey), the glutamate in the transmembrane helix of basigin (red), and the charged residues located in the negative (red) and positive patch (blue) of the Ig-I domain. The N-terminal signal peptide present in BSG var2 is shaded white. The expression constructs of this study were generated by truncations at the sites of changed grey levels between the extracellular domains (BSGΔIg, BSG Ig-I, BSG Ig I/C2, BSG var2); or lacked the basigin fusion altogether (MCT1). All contained an N-terminal hemagglutinin epitope-tag.
(PDF)

**S2 Fig. Delay of yeast growth during metabolic adaptation.** Jen1Δ ady2Δ yeast cultures with and without expression of MCT1 were switched from rich glucose conditions to liquid media containing 1% sodium lactate as the sole carbon source and growth was monitored by changes in the optical density (OD600).
(PDF)

**S3 Fig. Uptake of 14C-labeled L-lactate into jen1Δ ady2Δ yeast over time at pH 6.8 and a 1 mM inward gradient.** Shown are curves for BSG Ig-I/C2 E(27)R ($\bigtriangledown$) and BSG var2 E(27)R ($\triangle$). The data were normalized to 1 mg of cells and the background of non-expressing cells was subtracted. Error bars indicate ± S.E.M. from three biological replicates.
(PDF)

**S4 Fig. Michaelis-Menten kinetics for MCT1 fused with BSG variants.** Shown are curves for BSGΔIg ($\blacksquare$), BSG Ig-I/C2 ($\blacktriangledown$), BSG var2 ($\blacktriangle$). The data were normalized to 1 mg of cells and the background of non-expressing cells was subtracted. Error bars indicate ± S.E.M. from three biological replicates.
(PDF)

**S5 Fig. Western blot showing expression of MCT1 fusion constructs with BSG Ig-I carrying mutations of charged residues in the negative and positive patches.** Glu → Gln (Glu114,118,120,168,172), Lys/Arg → Ala (Lys108,111,127 plus Arg201,203) or as indicated.
(PDF)

**S6 Fig. Charged residues at the interface between the basigin Ig-I domain (BSG; R201 and R203) and the extracellular side of MCT1 (E46, K45, E48, E57, R175).** The structure data

are from PDB# 6lz0.
(PDF)

**S1 Table. Oligonucleotide list.**
(PDF)

**S1 Raw images.**
(PDF)

## Acknowledgments

We thank M. Casal for providing the *jen1Δ ady2Δ* yeast strain, C. Desel for confocal microscopy, U. Girreser for help with data evaluation, and the Institute of Clinical Molecular Biology in Kiel for providing Sanger sequencing as supported in part by the DFG Clusters of Excellence "Precision Medicine in Chronic Inflammation" and "ROOTS" with technical support by T. Naujoks, D. Langfeldt and B. Löscher. We further acknowledge A. Fuchs and B. Henke for excellent technical assistance, K. Geistlinger for discussions on the cell volume determination, and H. Anton-Beitz for language editing.

## Author Contributions

**Conceptualization:** Eric Beitz.

**Data curation:** Anna-Lena Köpnick, Annika Jansen, Katharina Geistlinger, Nathan Hugo Epalle, Eric Beitz.

**Formal analysis:** Anna-Lena Köpnick, Annika Jansen, Katharina Geistlinger, Nathan Hugo Epalle, Eric Beitz.

**Funding acquisition:** Eric Beitz.

**Investigation:** Anna-Lena Köpnick, Annika Jansen, Katharina Geistlinger, Nathan Hugo Epalle, Eric Beitz.

**Project administration:** Eric Beitz.

**Supervision:** Eric Beitz.

**Validation:** Anna-Lena Köpnick, Annika Jansen, Katharina Geistlinger, Nathan Hugo Epalle, Eric Beitz.

**Writing – original draft:** Anna-Lena Köpnick.

**Writing – review & editing:** Eric Beitz.

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
