## [Decision Letter · Decision Letter 0]

23 Feb 2021

PONE-D-21-03063

Basigin drives intracellular accumulation of l-lactate by harvesting protons and substrate anions

PLOS ONE

Dear Prof. Beitz,

Two Reviewers have carefully considered your manuscript.

In general, your manuscript has merit and addresses an interesting and exciting new potential function of basigin. Therefore, I invite you to submit a revised version of the manuscript that addresses the points raised by the two Reviewers. Reviewer 1 has several questions and suggestions that will clearly help strengthen the manuscript. Reviewer 2 has also very good points that will improve the manuscript, and specifically asks to discuss/explain how H+ and lactate can be channeled from the Ig-I antenna to MCT1.

A rebuttal letter that responds to each point raised by the two Reviewers. You should upload this letter as a separate file labeled 'Response to Reviewers'.A marked-up copy of your manuscript that highlights changes made to the original version. You should upload this as a separate file labeled 'Revised Manuscript with Track Changes'.An unmarked version of your revised paper without tracked changes. You should upload this as a separate file labeled 'Manuscript'.

We look forward to receiving your revised manuscript.

Kind regards,

Dimitrios Fotiadis

Academic Editor

PLOS ONE

Journal Requirements:

Reviewers' comments:

Reviewer's Responses to Questions

**Comments to the Author**

1. Is the manuscript technically sound, and do the data support the conclusions?

Reviewer #1: Yes

Reviewer #2: Yes

2. Has the statistical analysis been performed appropriately and rigorously? 

Reviewer #1: Yes

Reviewer #2: Yes

3. Have the authors made all data underlying the findings in their manuscript fully available?

Reviewer #1: Yes

Reviewer #2: Yes

4. Is the manuscript presented in an intelligible fashion and written in standard English?

Reviewer #1: Yes

Reviewer #2: Yes

5. Review Comments to the Author

Reviewer #1: Köpnick et al. report in their submitted manuscript PONE-D-21-03063 the role of the ancillary glycoprotein basigin on the intracellular accumulation of L-lactate.

Basigin associates with L-lactate-transporting members of the SLC16 family, which is also known as the monocarboxylate transporter (MCT) family, and mediates their trafficking to the plasma membrane. Basigin consists of one transmembrane helix, a long intracellular domain, as well as an extracellular domain, which is glycosylated and consists of different numbers of Ig-like domains (i.e. Ig-I, Ig-C2).

The authors established a S. cerevisiae-based assay, where they measure the uptake of radiolabeled L-lactate through concatemers consisting of MCT1 and versions of basigin (i.e., full-length and different truncations). Furthermore, the authors use the assay to monitor changes in the cytoplasmic pH upon exposure to L-lactate by using a pH-sensitive dye.

The authors show that the association of basigin with MCT1 leads to an intracellular accumulation of L-lactate, which is 4.5-fold above the substrate/proton concentrations provided by the external buffer.

The observed effect was assigned to the extracellular Ig-I domain of basigin by using basigin versions with truncated extracellular domains.

The authors identified surface patches of opposite (i.e., negative and positive) electrostatic potential on basigin, which they propose to act as proton and L-lactate binding sites, which locally increase substrate and co-substrate concentrations.

The authors conclude from their data that the Ig-I domain of basigin drives L-lactate uptake by locally increasing the proton and substrate concentrations at the extracellular side of MCT1.

Furthermore, the authors show that cell growth on media where L-lactate is the sole carbon source was strongly promoted in the presence of the Ig-I domain.

This manuscript is of interest for the readership of PLOS One and I recommend its publication after the following questions and concerns have been answered/addressed by the authors.

• The authors should provide line numbers in a revised version of the manuscript, which makes the review process much easier.

• The authors established a new transport assay using basigin-MCT1 concatemers. To my knowledge this has not been done before with any other member of the SLC16 family. Therefore, the authors should verify that the concatemers “behave” as expected by providing transport inhibition data using known and potent MCT1 inhibitors (e.g., AZD3965) and by providing kinetic data (i.e., Km-values and Vmax). Furthermore, it would be beneficial to provide transport data of the Arg313Ala, which should be non-functional.

• The authors must also cite the following publication: Grollman et al. Biochemistry 2000, 39, 9351, which describes the expression and functional characterization of MCT3 in S. cerevisiae. This could be done on page 5: [25,26].

• From mammalian cells it is known that basigin is glycosylated. Did the authors also observe glycosylation of basigin when concatenated to MCT1 and/or when expressed alone? If multiple bands are detected on Western blots, sample treatment with de-glycosidases might be insightful.

• When comparing confocal microscopy images of control cells with images of yeast containing expressed MCT1 or concatemers (Fig. 1c and 2b), one can detect fluorescent circles within the yeast cells. Can the authors comment on the origin of these circular objects? Do they represent populations of MCT1 or concatemers, which are stuck in intracellular vesicles?

• In Fig. 5 the authors used the X-ray structure of the Ig-domains of basigin (i.e., BSG var2) to calculate the electrostatic surface potential and to detect charged clusters. However, in light of the now available cryo-EM structure of the MCT1-basigin heterodimer I strongly recommend computing the electrostatic surface potential of the heterodimer in order to check if the identified charged clusters have counter charges on the MCT1 transporter.

• Although the authors focus on the role of basigin, they should also mention that negatively-charged aspartate residues on MCTs have also been reported to play important roles in the transport activity and pH dependency. They might function as a first step of substrate and proton recognition: Yamaguchi et al. Biochem Biophys Res Commun. 2020 Sep 3;529(4):1061-1065

• On p. 4, the authors must provide a reference for their statement that basigin homologs are absent in yeast. How is MCT1 then trafficked to the plasma membrane?

• Did the authors compare the expression levels of GFP- and non-GFP-tagged MCT1? Do they share common transport characteristics? Why did the authors not use the GFP-tagged version also for transport measurements to allow direct comparison with confocal data?

• On p. 6 the authors claim that the BSG∆Ig E27R-MCT1 fusion protein is well expressed and reached the plasma membrane. However, it does not mediate L-lactate transport. Do the authors conclude that E27 is directly important for L-lactate transport activity?

• For all presented constructs, the authors should provide membrane localization data in Table 1.

• In Table 1 the authors should provide Km and Vmax values for the presented constructs.

• In general, the authors should check their manuscript for consistency: total uptake or uptake capacity etc.

• On p. 6, the authors use the term “abscissa”. Do the authors mean the y- or x-axis?

• On p. 7, the authors mention that the Western blot indicates some protein fragmentation of the BSG Ig-I C23S and C82S constructs (Fig. 2F). This has also been observed for the normal Ig-I and Ig-I/C2 (Fig. 2A). The authors should perform Western blot analysis using antibodies against the basigin to check the identity of the observed bands.

• On p. 7, the authors claim that Cys-to-Ser mutations in the Ig-I domain render them misfolded. Is there any literature on the effect of disrupting the disulfide-bridges of the Ig-domains of basigin or related proteins? How can misfolded trafficking chaperones (i.e., basigin) mediate the proper trafficking of MCT1 to the plasma membrane? The authors must provide confocal microscopy data to show where the MCT1- BSG Ig-I C23S and C82S concatemers are located in the yeast. The authors should also measure L-lactate transport under reducing conditions, which also leads to breaking of the disulfide-bridges.

• On p. 8, the authors should give a reference for “8.6 x 10^6 cells per mg”.

• On p. 9, the authors mention “that the system was equilibrated”. However, this statement is only correct, if one assumes the highest measured intracellular L-lactate concentration of 0.44 mM. In the case of 0.18 mM, which was also measured, the system is not equilibrated.

• On p. 12; last line, the authors should modify their statement “by a factor of four” since the range of the factor is 0.3 – 3.5 according to the cited publication. Are the observed effects still virtually equal?

• On p. 13, I do not fully agree with the comparison of the here presented results with the effect of proteolytically modified basigin on intracellular lactate levels in lung cancer cells (ref. 16). In ref. 16 MCT4-mediated proton-coupled L-lactate efflux is studied whereas proton-coupled L-lactate uptake is studied in the here presented manuscript. MCT4-mediated proton-coupled L-lactate efflux requires an intracellular proton-binding site. However, in ref. 16 the mentioned protease functions on the extracellular side of basigin, which might not have anything to do with the intracellular proton binding.

• On p. 13, the authors should be more precise with the expression “L-lactate-releasing ….. cancers”. Do they mean cancer cells?

• On p. 14, there is a missing word in “The MCT1 ORF was cloned into using ….”

• Throughout the manuscript abbreviations – also of media etc. – should be introduced.

• On p. 15, replace “4.000 g” by “4,000 g”.

• On p. 15, please specify “twelve cycles”.

• On p. 15, please specify if the SDS-PAGE was done under reducing or non-reducing conditions.

• On p.16 and elsewhere, specify how the yeast cells were harvested (i.e., centrifugation speed etc.).

• On p. 17, was the uptake done under shaking and at which temperature?

• On p. 17, the authors mention “Michaelis-Menten kinetics”. However, no Km or Vmax values are reported in the manuscript. As far as I know, the specific activity is kept constant in Michaelis-Menten kinetics experiments (i.e., saturation curves), and not the amount of radioactivity as mentioned here: “final L-lactate concentrations of up to 10 mM were used each containing 0.1 uCi 14C-L-lactate”.

• On p. 17, the authors should mention for how long the scintillation counting was done.

• On p. 17, the authors should mention in the “Substrate transport assays” section how they converted the raw data (i.e., counts-per-minute) into “nmol”. Furthermore, it would be informative to mention the signal strength of a typical uptake experiment (i.e., counts-per-minute of untransformed yeast and MCT1-expressing yeast).

• On p. 17, the authors should provide a control experiment showing that the pH-sensitive dye remained inside the cell after loading.

• On p. 17, the authors should mention at which temperature the intracellular pH-measurements were done.

• On p. 18, the authors should mention the setting that were used for the Poisson-Boltzman electrostatics determination.

• On p. 18, the authors should specify for what they used the exponential fit (time-course?). What did they extract from the fit?

• Table 1: I recommend to provide amino acid numbers of all mutated residues.

• Fig 4A: The lines connecting the data points do not look like a fit while time-course experiments in other figures were always fitted using the described exponential model curve.

Reviewer #2: The manuscript by Köpnick and colleagues sustains that the extracellular Ig1 domain of Basigin acts as an antenna that harvest lactate- and H+ to generate a local concentration of these substrates that increments the intracellular by MCT1 transporter. This accumulation of substrates would perturb the gradient across the plasma membrane (Le Chatelier’s principle) resulting in a higher accumulation by MCT1 that the one obtained with MCT1 alone. Deletions and point mutations in the extracellular domain identify specific residues that are responsible of the antenna function of the IG1 domain of Basigin. The concept it is very interesting and gives to the ancillary protein Basigin a new functional role in lactate transport via MCT1. I have one concern about how the harvested H+ and lactate molecules are channelled to MCT1.

Major

1) Available structures of MCT1-Basigin complex with MCT1 in outward-facing conformations (e.g., PDB ID: 6LYY) shows that the residues responsible for harvesting H+ and lactate are far away from the substrate vestibule of MCT1. Indeed, Fig. 6 depicts this clearly by evoking an increased H+ and lactate concentration between the antenna and MCT1. In this situation how could be explained the channelling of the bound H+ and lactate from Ig-I to MCT1? If there is not an efficient channelling how could be explained that the H+ and lactate bound to the Ig-I antenna is not dissociated to the surrounding media according with the association/dissociation parameter (Kd) ? Are there residues in MCT1 vestibule that might channel H+ and lactate from the Ig-I antenna to the substrate binding site of MCT1? The authors should discuss/explain this point.

2) The experiments at pH 6.8 (4.5-fold accumulation of lactate) and 5.8 (2-fold accumulation) indicates that the influence of BSG Ig-I on lactate/H+ co-transport is higher when protons are scarce. Intuitively one could think that the H+ and lactate antenna Ig1 would decrease Km for co-transport of H+ and lactate. Is this the case?

Minor

1) Why Basigin mutant E27R (fusion protein BSG ΔIg E27R)abolishes transport without affecting surface expression. Please, clarify. This is not evident for a reader not introduced in MCTs.

Looks strange that using the N-terminal peptide signal of BSG (BSG var2) there is lower total and surface expression of the protein. Any explanation?

6. PLOS authors have the option to publish the peer review history of their article (what does this mean?). If published, this will include your full peer review and any attached files.

Reviewer #1: No

Reviewer #2: **Yes: **Manuel Palacín

---

## [Author Response · Author response to Decision Letter 0]

6 Mar 2021

Dear Dr. Fotiadis,

We appreciate the insightful and comprehensive comments by the reviewers. To address the points raised, we added new data (Km, vmax), Figures, and text passages, we rephrased and clarified wording, and included new references as suggested. The changes greatly added value to the manuscript, and we feel confident that the revised version has gained clarity in the sense of the reviewers.

In the following, we address the reviewer’s comments on a point-by-point basis:

Reviewer 1

• The authors should provide line numbers in a revised version of the manuscript, which makes the review process much easier.

Done, line number were added.

• The authors established a new transport assay using basigin-MCT1 concatemers. To my knowledge this has not been done before with any other member of the SLC16 family. Therefore, the authors should verify that the concatemers "behave" as expected by providing transport inhibition data using known and potent MCT1 inhibitors (e.g., AZD3965) and by providing kinetic data (i.e., Km-values and Vmax). Furthermore, it would be beneficial to provide transport data of the Arg313Ala, which should be non-functional.

We added Km and vmax data of the fused and unfused MCT1 constructs as suggested. The obtained values correspond to previously published data, and are displayed in two new Figures (5F, and S4) as well in Table 1; added text in lines 209-220. We have further done Arg313 mutations of the unfused construct that rendered the transporter non-functional, and carried out inhibitor assays with the Astra Zeneca compound. The latter also behaved as expected and yielded IC50 values around 100 nM for both, fused and unfused MCT1. We refrain from including this data in the revision, because it is already included in another manuscript that is under review at another journal.

• The authors must also cite the following publication: Grollman et al. Biochemistry 2000, 39, 9351, which describes the expression and functional characterization of MCT3 in S. cerevisiae. This could be done on page 5: [25,26].

Done: new reference [28] (line 115)

• From mammalian cells it is known that basigin is glycosylated. Did the authors also observe glycosylation of basigin when concatenated to MCT1 and/or when expressed alone? If multiple bands are detected on Western blots, sample treatment with de-glycosidases might be insightful.

Our experience with transporter expression in the yeast system shows that the proteins typically remain unglycosylated and our Western blots are not indicative of multiple bands that would refer to glycosylation. We provide this statement and several references on this in the ms (lines 165-168).

• When comparing confocal microscopy images of control cells with images of yeast containing expressed MCT1 or concatemers (Fig. 1c and 2b), one can detect fluorescent circles within the yeast cells. Can the authors comment on the origin of these circular objects? Do they represent populations of MCT1 or concatemers, which are stuck in intracellular vesicles?

The intracellular compartment in which the excess MCT1 constructs reside are typically perinuclear and the yeast vacuole membranes. Such protein does not interfere with transport activity at the plasma membrane. This is explained in the new lines 118-120.

• In Fig. 5 the authors used the X-ray structure of the Ig-domains of basigin (i.e., BSG var2) to calculate the electrostatic surface potential and to detect charged clusters. However, in light of the now available cryo-EM structure of the MCT1-basigin heterodimer I strongly recommend computing the electrostatic surface potential of the heterodimer in order to check if the identified charged clusters have counter charges on the MCT1 transporter.

Unfortunately, the resolution of the cryo-EM structure in the area of the charged residues of the basigin Ig-I domain is too low. In fact, the sidechains of the relevant residues are missing in the structure (only Arg201 and Arg203 are visible). Therefore, the electrostatics calculation must be done with the high-resolution crystal structure (stated in the ms, lines 262-265). However, we included a new Figure S6 showing that BSG Ig-I Arg201 and Arg203 interface with a cluster of five charged residues at the MCT1 extracellular face (Lys45, Glu46, Glu48, Glu57, Arg175). Here, indeed proton/substrate transfers may be facilitated. We are grateful for the comment as this addition appears quite valuable.

• Although the authors focus on the role of basigin, they should also mention that negatively-charged aspartate residues on MCTs have also been reported to play important roles in the transport activity and pH dependency. They might function as a first step of substrate and proton recognition: Yamaguchi et al. Biochem Biophys Res Commun. 2020 Sep 3;529(4):1061-1065

We added the information (lines 332-334) and the reference as [45]; in light of the point above, this is certainly a highly valuable suggestion.

• On p. 4, the authors must provide a reference for their statement that basigin homologs are absent in yeast. How is MCT1 then trafficked to the plasma membrane?

We provide a new reference [25] to the yeast genome database that contains the information.

• Did the authors compare the expression levels of GFP- and non-GFP-tagged MCT1? Do they share common transport characteristics? Why did the authors not use the GFP-tagged version also for transport measurements to allow direct comparison with confocal data?

We wanted to keep the MCT1 transporter as close to the original sequence as possible; this is also valid for the fusion constructs with basigin because no additional linker was introduced between both partners. The reviewer is correct in stating that this may affect the correlation of transport data with the microscopy imaging. However, since the focus of the study is on the equilibrium transport state in which the actual number of transporters at the plasma membrane is irrelevant, we prefer to keep the measurements as done with the untagged transport protein.

• On p. 6 the authors claim that the BSG∆Ig E27R-MCT1 fusion protein is well expressed and reached the plasma membrane. However, it does not mediate L-lactate transport. Do the authors conclude that E27 is directly important for L-lactate transport activity?

No, we assume that disruption of the interaction between basigin and MCT1 in the membrane domain at a sustained close link of the fused proteins interferes with a proper arrangement of the transmembrane spans leading to a loss of transport functionality. This is now stated in the text (lines 154-157).

• For all presented constructs, the authors should provide membrane localization data in Table 1.

As pointed out above, the study focuses on the transport equilibrium state in which the actual number of transporters at the plasma membrane is irrelevant. Accordingly, functionality of a construct is sufficient to show plasma membrane localization. Therefore, microscopy data is provided only for the initial establishment of the fusion constructs.

• In Table 1 the authors should provide Km and Vmax values for the presented constructs.

We added Km and vmax for selected constructs as suggested (new Fig. 5F, Table 1) and added a new paragraph in the text (lines 209-220).

• In general, the authors should check their manuscript for consistency: total uptake or uptake capacity etc.

Changed to “capacity”.

• On p. 6, the authors use the term "abscissa". Do the authors mean the y- or x-axis?

Changed to y-axis

• On p. 7, the authors mention that the Western blot indicates some protein fragmentation of the BSG Ig-I C23S and C82S constructs (Fig. 2F). This has also been observed for the normal Ig-I and Ig-I/C2 (Fig. 2A). The authors should perform Western blot analysis using antibodies against the basigin to check the identity of the observed bands.

The reviewer is referring to the approx. 25 kDa fragment that is visible in the blots. The fragment derives from minimal breakage of the MCT1 protein in the region of the large intracellular loop. We observed this for virtually all MCT isoforms that we expressed in the yeast. The fused basigin domain is of about the same size, hence the appearance of a band at the apparent size of MCT1 alone for some fusion constructs predominantly for the C-to-S mutants which probably due to a certain degree of misfolding appear to particularly prone to fragmentation. We think that the fragments can be attributed very well from our blots and further investigation would not gain critical improvement especially in light of the clear functional data. Therefore, we would like to refrain from purchasing a basigin antibody for additional experiments in this case.

• On p. 7, the authors claim that Cys-to-Ser mutations in the Ig-I domain render them misfolded. Is there any literature on the effect of disrupting the disulfide-bridges of the Ig-domains of basigin or related proteins? How can misfolded trafficking chaperones (i.e., basigin) mediate the proper trafficking of MCT1 to the plasma membrane? The authors must provide confocal microscopy data to show where the MCT1- BSG Ig-I C23S and C82S concatemers are located in the yeast. The authors should also measure L-lactate transport under reducing conditions, which also leads to breaking of the disulfide-bridges.

Again, the functional data in combination with the cellular localization via microscopy clearly suggest this interpretation. There is no other literature, yet, on the modification of the disulfide bridges of basigin that we could cite. The trafficking machinery in the yeast differs greatly from that of mammalian cells especially in terms of signal peptides. In a heterologous yeast overexpression system, a certain proportion of the protein makes it into the membrane; this is also the case for the Xenopus oocyte system where MCTs (and many other human membrane proteins) reach the plasma membrane without the human trafficking machinery present. Expression and functional characterization of human membrane proteins in a different species such as yeast is well established over decades and only possible due to this type of overloading and circumventing the quality control and trafficking procedures.

• On p. 8, the authors should give a reference for "8.6 x 10^6 cells per mg".

Done, we cite ref [31] of the end of last year in which we determined this number.

• On p. 9, the authors mention "that the system was equilibrated". However, this statement is only correct, if one assumes the highest measured intracellular L-lactate concentration of 0.44 mM. In the case of 0.18 mM, which was also measured, the system is not equilibrated.

We changed the wording in lines 237-240. Three out of four constructs yielded values of 0.39-0.44 mM, i.e. equal within the error margin. With the gradient factor of 2.5 this indicates an equilibrated system. Only the weakly functional BSG Ig-I C82S mutant exhibited the lower 0.18 mM value. We point this out in the text.

• On p. 12; last line, the authors should modify their statement "by a factor of four" since the range of the factor is 0.3 – 3.5 according to the cited publication. Are the observed effects still virtually equal?

We adapted the text (lines 358-359) to indicate the maximal factor of 3.5; the way the referenced authors derived their data is very different from our assay system. Yet, the conditions at which their factor of 3.5 was obtained justify relating the numbers.

• On p. 13, I do not fully agree with the comparison of the here presented results with the effect of proteolytically modified basigin on intracellular lactate levels in lung cancer cells (ref. 16). In ref. 16 MCT4-mediated proton-coupled L-lactate efflux is studied whereas proton-coupled L-lactate uptake is studied in the here presented manuscript. MCT4-mediated proton-coupled L-lactate efflux requires an intracellular proton-binding site. However, in ref. 16 the mentioned protease functions on the extracellular side of basigin, which might not have anything to do with the intracellular proton binding.

It is quite possible that additional effects from individual proton or substrate attracting residues in MCT1 and MCT4 contribute here. At the core of our discussion lies the hypothesis that the basigin Ig-I domain creates a microenvironment at the extracellular face of the MCT that locally increases the available proton and substrate concentration. In this case, the classical equilibrium principle would explain the shifted substrate distribution between the intracellular and extracellular space. This hypothesis is certainly new to the field. Whether this holds true for MCT4 as well requires a new study in the future. For now, we would like to indicate that MCT1 (tested here) and MCT4 (data from the literature) may behave comparably.

• On p. 13, the authors should be more precise with the expression "L-lactate-releasing ..... cancers". Do they mean cancer cells?

Done, changed to “cancer cells” (line 376).

• On p. 14, there is a missing word in "The MCT1 ORF was cloned into using ...."

Done, the missing plasmid name “pDR196” was added (line 397).

• Throughout the manuscript abbreviations – also of media etc. – should be introduced.

We completed the abbreviations list to include the immunoglobulin-like domains and the synthetic dropout media (lines 38-40).

• On p. 15, replace "4.000 g" by "4,000 g".

Corrected.

• On p. 15, please specify "twelve cycles".

We eliminated “cycles” and rephrased the text to better explain the twelve repetitions of the procedure (lines 433-434).

• On p. 15, please specify if the SDS-PAGE was done under reducing or non-reducing conditions.

Reducing conditions added in line 439.

• On p.16 and elsewhere, specify how the yeast cells were harvested (i.e., centrifugation speed etc.).

The harvesting parameters (4,000 g, 5 min, 4 °C) were added (line 467).

• On p. 17, was the uptake done under shaking and at which temperature?

The temperature (19 °C) and non-shaking conditions were added (line 473)

• On p. 17, the authors mention "Michaelis-Menten kinetics". However, no Km or Vmax values are reported in the manuscript. As far as I know, the specific activity is kept constant in Michaelis-Menten kinetics experiments (i.e., saturation curves), and not the amount of radioactivity as mentioned here: "final L-lactate concentrations of up to 10 mM were used each containing 0.1 uCi 14C-L-lactate".

The Km and vmax values were added to the paper (new Figures, extended Table, text; see above). The concentration of radiolabeled lactate (µM range) in relation the “cold” lactate (mM) is negligible. The 14C-label is used as a “spike” that is relative to the total amount of lactate in the assay; the equation to calculate the cytosolic lactate takes this into account, see two points below (and new lines 482-483 in the paper).

• On p. 17, the authors should mention for how long the scintillation counting was done.

“2 min” added (line 482).

• On p. 17, the authors should mention in the "Substrate transport assays" section how they converted the raw data (i.e., counts-per-minute) into "nmol". Furthermore, it would be informative to mention the signal strength of a typical uptake experiment (i.e., counts-per-minute of untransformed yeast and MCT1-expressing yeast).

Both, the equation and the range of cpm counts were added (lines 482-485).

• On p. 17, the authors should provide a control experiment showing that the pH-sensitive dye remained inside the cell after loading.

We refer to our ref [31] in which we established the method for our measurements and added wording to the text (lines 493-494).

• On p. 17, the authors should mention at which temperature the intracellular pH-measurements were done.

“Room temperature” was added (line 502).

• On p. 18, the authors should mention the setting that were used for the Poisson-Boltzman electrostatics determination.

The settings were added: pH 7.0, protonation state assignment by PROPKA, and PARSE forcefield (lines 511-512).

• On p. 18, the authors should specify for what they used the exponential fit (time-course?). What did they extract from the fit?

“time-course” added as well as the calculation procedure: transport rates from the slope at t = 0, and uptake capacities from the upper limit of each curve (lim┬(t→∞)⁡〖f(t)〗=a) (lines 516-519).

• Table 1: I recommend to provide amino acid numbers of all mutated residues.

The numbers were added to the Table (and respective text passages, and Figure legends).

• Fig 4A: The lines connecting the data points do not look like a fit while time-course experiments in other figures were always fitted using the described exponential model curve.

We aimed at reaching the plateau, the time-course is not relevant here. Therefore, a limited number of data points focusing at the late, equilibrium state were determined that prohibited a fair fitting.

Reviewer 2

Major

1) Available structures of MCT1-Basigin complex with MCT1 in outward-facing conformations (e.g., PDB ID: 6LYY) shows that the residues responsible for harvesting H+ and lactate are far away from the substrate vestibule of MCT1. Indeed, Fig. 6 depicts this clearly by evoking an increased H+ and lactate concentration between the antenna and MCT1. In this situation how could be explained the channelling of the bound H+ and lactate from Ig-I to MCT1? If there is not an efficient channelling how could be explained that the H+ and lactate bound to the Ig-I antenna is not dissociated to the surrounding media according with the association/dissociation parameter (Kd) ? Are there residues in MCT1 vestibule that might channel H+ and lactate from the Ig-I antenna to the substrate binding site of MCT1? The authors should discuss/explain this point.

This point was also raised by Reviewer 1. For the electrostatics calculation, the new structure, unfortunately, is not usable because most of the relevant amino acid sidechains in the basigin Ig-I domain are not resolved and actually missing the structure data. Yet, Arg201 and Arg203 are there, and, indeed when we look at the interface with MCT1 there is a cluster of charged residues present that may act to shuttle protons or substrate into the transporter. We have added a new Figure S6 showing the interface and a respective passage in the discussion. In this regard, the proton antenna function of the carbonic anhydrase, which is even farther away from the transporter, may have to be viewed differently, probably with basigin Ig-I acting as a “bridge” in the translocation. This view of the cryo-EM structure is truly enlightening and adds value to the paper.

2) The experiments at pH 6.8 (4.5-fold accumulation of lactate) and 5.8 (2-fold accumulation) indicates that the influence of BSG Ig-I on lactate/H+ co-transport is higher when protons are scarce. Intuitively one could think that the H+ and lactate antenna Ig1 would decrease Km for co-transport of H+ and lactate. Is this the case?

We added Km and vmax data (see also points by Reviewer 1). In fact, the addition of basigin increased the Km which might either indicate an apparent lower substrate affinity or it may reflect the proposed disturbance of the proton/substrate concentration distribution between the buffer and the local microenviroment established between the basigin Ig-I domain and MCT1. A locally increased availability of the substrate and co-substrate would relate only indirectly to the overall buffer concentration (which is the basis for the Km calculation). This effect is most likely pH-dependent and the relation is not necessarily proportional to the buffer pH.

Minor

1) Why Basigin mutant E27R (fusion protein BSG ΔIg E27R)abolishes transport without affecting surface expression. Please, clarify. This is not evident for a reader not introduced in MCTs.

We assume that disruption of the interaction between basigin and MCT1 in the membrane domain at a sustained close link of the fused proteins interferes with a proper arrangement of the transmembrane spans leading to a loss of transport functionality. This is now stated in the text (lines 154-157).

Looks strange that using the N-terminal peptide signal of BSG (BSG var2) there is lower total and surface expression of the protein. Any explanation?

Since we express a human protein in the yeast system, the trafficking machinery is not working in the classical way. It is overwhelmed by the large amount of overexpressed protein and a certain proportion of it reaches the plasma membrane (a considerable amount is retained in intracellular compartments, though; see microscopy images). Therefore, it is hard to predict what specifically hampers trafficking of the BSG var2 construct. The overall lower expression levels might not suffice to circumvent the quality control system of the yeast in a way the other constructs do. Anyway, the study focusses on the equilibrium state which is independent of the number of transporters at the plasma membrane; therefore, the overall data of this work remain interpretable.

In conclusion, we have addressed all points by the reviewers and feel confident that our major point of the study, i.e. creation of a local microenvironment between the basigin Ig-I domain and the extracellular face of MCT1, is corroborated by the changes made to the manuscript.

With this, we would appreciate if you now accept our paper for publication.

Best regards,

Eric Beitz

---

## [Decision Letter · Decision Letter 1]

12 Mar 2021

Basigin drives intracellular accumulation of l-lactate by harvesting protons and substrate anions

PONE-D-21-03063R1

Dear Prof. Beitz,

We’re pleased to inform you that your manuscript has been judged scientifically suitable for publication and will be formally accepted for publication once it meets all outstanding technical requirements.

Kind regards,

Prof. Dimitrios Fotiadis

Academic Editor

PLOS ONE

Reviewers' comments:

Reviewer's Responses to Questions

**Comments to the Author**

1. If the authors have adequately addressed your comments raised in a previous round of review and you feel that this manuscript is now acceptable for publication, you may indicate that here to bypass the “Comments to the Author” section, enter your conflict of interest statement in the “Confidential to Editor” section, and submit your "Accept" recommendation.

Reviewer #1: All comments have been addressed

2. Is the manuscript technically sound, and do the data support the conclusions?

Reviewer #1: Yes

3. Has the statistical analysis been performed appropriately and rigorously? 

Reviewer #1: Yes

4. Have the authors made all data underlying the findings in their manuscript fully available?

Reviewer #1: Yes

5. Is the manuscript presented in an intelligible fashion and written in standard English?

Reviewer #1: Yes

6. Review Comments to the Author

Reviewer #1: In the revised version of the manuscript PONE-D-21-03063R1, the authors have addressed all of my questions and concerns. Therefore, I recommend the acceptance of the revised version of the manuscript for publication in PLOS One.

7. PLOS authors have the option to publish the peer review history of their article (what does this mean?). If published, this will include your full peer review and any attached files.

Reviewer #1: No

---

## [Editor Report · Acceptance letter]

16 Mar 2021

PONE-D-21-03063R1 

Basigin drives intracellular accumulation of l-lactate by harvesting protons and substrate anions 

Dear Dr. Beitz:

I'm pleased to inform you that your manuscript has been deemed suitable for publication in PLOS ONE. Congratulations! Your manuscript is now with our production department. 

Kind regards, 

on behalf of

Prof. Dimitrios Fotiadis 

Academic Editor

PLOS ONE